# Learning a deep language model for microbiomes: The power of large scale unlabeled microbiome data

**Quintin Pope****[1]\***, **Rohan Varma[1]‡, Christine Tataru[2]‡, Maude M. David[3], Xiaoli Fern[1]**

**1** School of Electrical Engineering and Computer Science, Oregon State University, Corvallis, Oregon, United States of America, **2** Department of Pathology, Brigham and Women's Hospital, Boston, Massachusetts, United States of America, **3** Department of Pharmaceutical Sciences, Oregon State University, Corvallis, Oregon, United States of America

‡ Work performed as a student at Oregon State University.

\* popeq@oregonstate.edu

**Data availability statement:** American Gut Project, Halfvarson, and Schirmer raw data are available from the NCBI database (accession numbers PRJEB11419, PRJEB18471, and

## Abstract

We use open source human gut microbiome data to learn a microbial "language" model by adapting techniques from Natural Language Processing (NLP). Our microbial "language" model is trained in a self-supervised fashion (i.e., without additional external labels) to capture the interactions among different microbial taxa and the common compositional patterns in microbial communities. The learned model produces contextualized taxon representations that allow a single microbial taxon to be represented differently according to the specific microbial environment in which it appears. The model further provides a sample representation by collectively interpreting different microbial taxa in the sample and their interactions as a whole. We demonstrate that, while our sample representation performs comparably to baseline models in in-domain prediction tasks such as predicting Irritable Bowel Disease (IBD) and diet patterns, it significantly outperforms them when generalizing to test data from independent studies, even in the presence of substantial distribution shifts. Through a variety of analyses, we further show that the pretrained, context-sensitive embedding captures meaningful biological information, including taxonomic relationships, correlations with biological pathways, and relevance to IBD expression, despite the model never being explicitly exposed to such signals.

## Author summary

Human microbiomes and their interactions with various body systems have been linked to a wide range of diseases and lifestyle variables. To understand these links, citizen science projects such as the American Gut Project (AGP) have provided large opensource datasets for microbiome investigation. In this work we leverage such open-source data and learn a "language" model for human gut microbiomes using techniques derived

PRJNA398089, respectively). We used the curated data produced by Tataru and David, 2020 (https://doi.org/10.1371/journal.pcbi.1007859). All data and code required for our methods are made available and described in a Dryad repository at https://doi.org/10.5061/dryad.tb2rbp08p. Data and code available from: https://datadryad.org/stash/dataset/doi:10.5061/dryad.tb2rbp08p (data) and https://doi.org/10.5281/zenodo.13858903 (code). File descriptions and usage instructions are available in the repository's README.

**Funding:** This work was supported by the National Science Foundation Division of Emerging Frontiers (2025457 to XF and MD), as well as the Open Philanthropy Long-Term Future Scholarship Program (to QP). The funders had no role in study design, data collection and analysis, decision to publish, or preparation of the manuscript.

**Competing interests:** The authors have declared that no competing interests exist.

from natural language processing. We train the "language" model to capture the interactions among different microbial taxa and the common compositional patterns that shape gut microbiome communities. By considering the entirety of taxa within a sample and their interactions, our model produces a representation that enables contextualized interpretation of individual microbial taxon within their microbial environment. Despite their simple training signal, our contextualized sample representations distill broadly applicable biological information adaptable to multiple downstream tasks. We demonstrate that our sample representation enhances prediction performance compared to similar representation-learning baselines across multiple microbiome tasks including prediction of Irritable Bowel Disease (IBD) and diet patterns. Furthermore, our learned representation yields a robust IBD prediction model that generalizes well to independent data collected from different populations. Our in-depth analysis of the learned embeddings revealed that our pretrained model captured biologically meaningful information, despite never being explicitly exposed to such signals. Specifically, we found that the embeddings reflected taxonomic relationships in their geometry. Additionally, we observed significant correlations between the embedding dimensions and known metabolic pathways. Finally, sensitivity analysis of our IBD model highlights both known IBD-associated taxa and potentially novel taxa

# 1 Introduction

Identifiable features of the human microbiome and its interactions with various body systems have been associated with a wide range of diseases, including cancer [1], depression [2,3] and inflammatory bowel disease [4–6]. As our knowledge of such connections has advanced, research on the human microbiome has undergone a shift in focus, moving from establishing links to unraveling the underlying mechanisms and utilizing them to develop clinical interventions [7]. This transition has sparked interest in applying statistical methods to microbiome data, leading to the launch of open source projects such as the American Gut Project (AGP) and Human Food Project (HFP), which provide open source datasets for microbiome investigation [8]. These repositories offer data in the form of raw genetic reads, which, even after being processed into taxa counts, still present thousands of features per sample. Consequently, researchers often employ dimension reduction techniques to transform this data into a more manageable feature space.

Significantly, the relevance of microbes to any particular analysis is often intertwined with the presence and potential interactions of other microbes in the environment. However, common techniques for reducing microbiome data dimensions – such as binning based on phylogenetic relationships [9,10], clustering by gene similarity [11], or using PCA and other techniques [12] – don't account for the interactions between taxa when producing lower dimensional representations of samples. Consequently, a significant challenge in microbiome data analysis is to produce lower dimension representations (embeddings) of samples that not only take into account the presence of specific taxa but also their interactions and overall functioning as a whole.

Fortunately, a similar challenge has been investigated in the natural language processing (NLP) domain, which shares many similarities with the microbiome domain. Just as a sample comprises numerous microbes, a sentence consists of multiple words. Similarly, certain microbes hold greater relevance for specific analyses, while certain words are more important for different NLP tasks. Furthermore, just as a microbe can assume different functional roles under varying conditions, a word can possess different meanings in different contexts.

Given the strong similarities between the two domains and the shared goal of producing quality lower-dimensional sample/sentence representations, there is a growing interest in applying NLP techniques to microbiome analysis. Notably, previous work has successfully applied NLP word embedding algorithms to microbiome data, generating taxa embeddings that have shown promising results surpassing the performance of traditional dimension reduction techniques like PCA for various microbiome prediction tasks [13].

Specifically, [13] apply the GloVe (Global Vectors for Word Representation) embedding algorithm [14] to co-occurrence data derived from the AGP dataset. GloVe maps each taxon in the vocabulary to a vector representation, and optimizes those vectors such that the inner product of any two vectors will match the log of the co-occurrence rate of the associated pair of taxa.

However, this prior work [13] has several limitations. First, the embeddings are learned based on aggregated global microbe-to-microbe co-occurrence statistics—in reality, microbe interactions can be dynamic and context-dependent. Second, given a sample containing many taxa, the embedding for the sample is computed by taking an abundance-weighted-average of the taxa embeddings without considering the context-specific roles of individual microbes in the sample.

Similar to how the word "fly" changes from an insect in "I caught a fly" to an action in "I like to fly" based on context, the role of a bacteria can also shift based on its context and interactions. For example, susceptibility to infection with Campylobacter jejuni was shown to depend on the species composition of the microbiota [15].

Transformers, a powerful and flexible machine learning architecture originally developed for NLP [16], provides a potential solution to the above issues. Past work [17–21] has applied transformers to biological data. However, such work has focused on learning a sequence encoder for representing DNA [21] or, more commonly, protein amino acid sequences [17–20] (e.g., each token might represent a k-mer in such a sequence). In contrast, we focus on representing entire microbial communities and their interactions, using each token to represent a single microbe in such a community.

We present the first use of transformers to learn representations of microbiome at the taxa level by adapting "self-supervised" pre-training techniques from NLP, allowing the model to learn from vast amounts of unlabeled 16S microbiome data and mitigating the required amount of expensive labeled data. The pre-trained models can be viewed as a form of "language model" for microbiome data, capturing the inherent composition rules of microbial communities, which we can easily adapt to downstream prediction tasks with a smaller amount of labeled "finetuning" data.

We show that using a transformer model pre-trained on data from the American Gut Project (AGP) as the starting point, we can learn representations that capture biologically meaningful patterns that we can repurpose for multiple downstream tasks, as supported by multiple complementary experiments. We find our method surpasses wide-ranging representation learning baselines in performance for multiple downstream host phenotype prediction tasks including IBD disease state prediction. We show state of the art cross-study and dataset generalization performance. We find strong association between the geometric structure of our representations and the taxonomic relationships among ASVs. We also show unmatched correlation between contextualized representation embedding dimensions and known metabolic pathways. Finally, we investigate the most significant taxa that drive IBD classification predictions and find connections to known drivers of gut health and disease states. These results showcase the remarkable capability of the pre-trained microbial "language" model in generating enhanced representation of the microbiome.

Focusing on the IBD prediction task, we demonstrate that our IBD prediction model, trained on the IBD data from the American Gut Project, with a simple ensemble strategy, exhibits robust generalization across several IBD studies with notable distributional shifts. We further visualize the contextualized taxa embeddings produced by our pre-trained language model and show that they capture biologically meaningful information, both taxonomic associations between ASVs as well as known metabolic pathways. Finally, we analyze the learned IBD prediction model to identify taxa that strongly influence the model's prediction.

## 2 Materials and methods

We begin by introducing the general workflow of applying a transformer model for generating a sample embedding (Fig 1) and explaining each step of the work flow, including a detailed look into the transformer architecture. We then explain how we perform the pretraining, followed by finetuning for specific down stream tasks (Fig 2). This section will also explain how we identify those taxa that most affect the model's classification decisions (Eq 1) and conclude with a description of the datasets used in this paper.

### 2.1 Transformers for microbiome data: Workflow overview

Since their introduction in 2017, [16] transformers have emerged as one of the most powerful classes of neural models invented to date, demonstrating state-of-the-art performance in many domains, though different tasks and data types require specific adaptations. Fig 1 summarizes the basic workflow of applying transformer to microbiome data for generating sample representations and context-sensitive taxa embeddings.

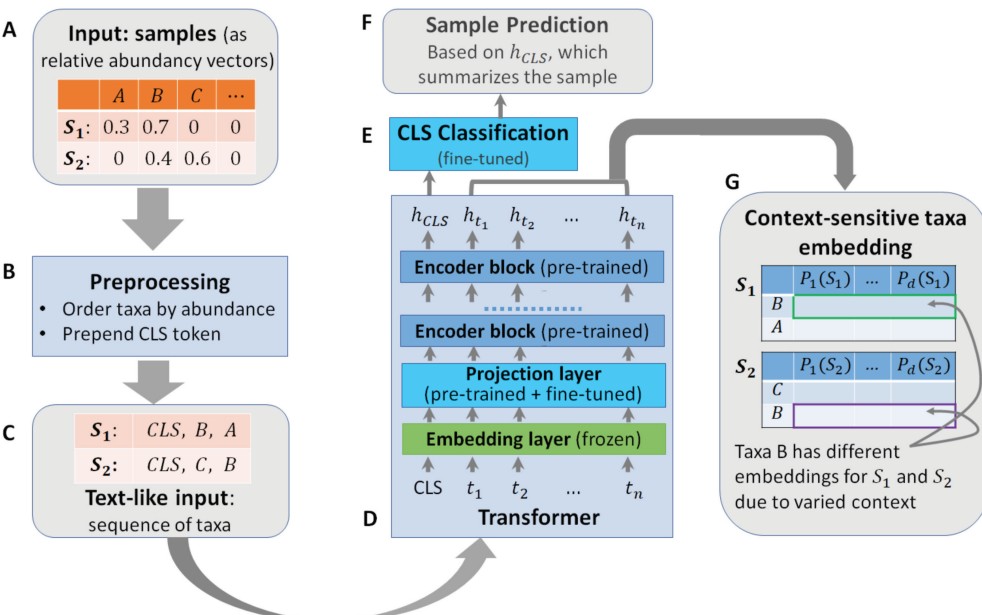

**Fig 1. Workflow of using a transformer model for generating sample embedding/classification and context sensitive taxa embeddings.** The inputs (A), which are samples represented as relative abundance vectors, first go through the preprocessing step (B) to generate text-like inputs (C) for the transformer model (D). The transformer model generates a sample embedding ($h_{cls}$) that goes through a sample classification layer (E) to produce task specific sample level predictions (F). The transformer model also generates context sensitive embedding (G) for each taxa in the sample. The same taxa appearing in different samples can have different embedding because of contextual differences.

**Transformer Training Process**

**Fig 2. Training of the transformer model.** Unlabeled microbiome data (A) is fed into a randomly initialized transformer (B) as inputs to the self-supervised pre-training process (C), which produces a pre-trained transformer that generates token-level classifications (D). We replace the token-level classification head with a randomly initialized CLS classification head (E), and use labeled microbiome data (F) to fine-tune the CLS classification head (G), which produces the fine-tuned transformer (H).

**Preprocessing steps.** We assume that microbiome samples are represented as vectors of relative taxa abundances (Fig 1A). To prepare our input for the transformer model, we perform a pre-process step (Fig 1B) to transform the microbiome sample into 'text-like' inputs (Fig 1C). Specifically, we rank all the taxa present in the sample in decreasing order of abundance to create an ordered list of taxa (truncated to contain no more than the 512 most abundant taxa). This step creates inputs that are analogous to texts, which are ordered lists of tokens of variable length capped at 512. Transformer computational costs increase with the square of their input lengths, so truncating inputs to at most 512 helps ensure our method remains computationally efficient to run, while affecting less than 6% of the training data points.

Similar to what is done in processing textual inputs, we prepend a special 'classification (CLS)' token to our input list. We use the 'CLS' token's representation as the final sample representation, which we treat as a summary of the full sample for classification purposes.

**The transformer model.** Fig 1D provides a sketch of our transformer architecture for performing a sample classification task. The input to the transformer model is an ordered list of taxa. The list first goes through an embedding layer and a projection layer. The output of the projection layer then feeds into a sequence of multiple encoder blocks (we use 5 encoder blocks in this work), where each encoder block produces a new representation based on outputs of the previous block. Below we explain the individual components.

*Embedding layer.*

The embedding layer maps from discrete tokens/taxa to their corresponding vector representations. We use absolute positional embeddings [16] to encode the abundance-based taxa order into the taxa embeddings. We experimented with a variety of methods to incorporate abundance information, including different positional embedding methods such as relative key [22] and relative key query [23] methods, as well as using additional embedding dimensions to directly store abundance values. We found little difference between these methods,

and hence opted for the absolute positional embeddings based on rank ordering for its relative simplicity.

We preset the embedding layer using the 100-dimensional GloVe taxa embedding from [13], learned using the co-occurrence data from the AGP dataset, and keep it frozen during training, except for the 'CLS' token embedding, which is initialized randomly and trained during pre-training and fine-tuning. We do this to enable a more direct comparison of the contextualized embeddings with the original vocabulary embedding learned through GloVe, thus emphasizing the benefits of contextualized representations.

*Projection layer.*

The projection layer is a linear transformation from the vocabulary embedding space to the model's hidden representation space. The projection layer allows the model to process inputs of different dimensionality than the model's hidden space. In this work, the projection layer projects from the 100 dimensional vocabulary embedding into a richer 200 dimensional hidden space used by the model.

*Encoder blocks.*

This is where the transformer begins incorporating "context" into the representation of each ASV in the sample. Here we provide an intuitive explanation of the encoder block. Please see [16] for concrete mathematical definitions.

An encoder block consists of a multi-headed self-attention layer [16] and a fully connected layer. The multi-headed attention layer computes a set of self-attention scores (one per head). Each attention head can read and write to different subspaces in the embeddings, and can track its own set of all-pairs interactions between every taxon in the sample. This could allow different heads to track different collections of statistical factors that influence community composition and metabolic functions.

The network modulates how much 'attention' is paid to each context taxon when updating the representation for a particular taxon in the sample. For example, in the context of language and given a sentence such as "I waved at the band, but they didn't see me", a properly trained encoder block could update the embedding of word "they" to reflect that it is referencing "the band". Analogously, in microbiome data, if bacteria *A* performs a functional role conditioned on the presence of bacteria *B*, a properly trained encoder block could update the embedding for bacteria *A* to reflect the presence or absence of bacteria *B*.

*Classification head.*

We rely on a special 'CLS' token to summarize information from all the other taxa/tokens. The CLS token then feeds into a classification head, which is a standard two-layer feed-forward neural network with 200 hidden nodes, to produce a prediction for a specific classification task.

## 2.2 Transformer training

A critical challenge in applying complex deep learning models like transformers is the lack of large amounts of labeled training data. This can be addressed, however, using a technique referred to as self-supervised pre-training [24], which leverages readily available unlabeled data. In this work, we follow this approach and our training process is described in Fig 2.

**Pre-training.** We begin with a randomly initialized transformer and first train a task-agnostic transformer using unlabeled data via self-supervised pre-training. Specifically, We use ELECTRA (Efficiently Learning an Encoder that Classifies Token Replacements Accurately) [25] to pre-train the encoder layers of the transformer model. We chose ELECTRA

because it reaches comparable performance to other popular pre-training approaches (BERT [26] and its various flavors) while being computationally efficient.

The ELECTRA pre-training approach has two steps. The first step trains a generator model by randomly masking out 15% of taxa in microbiome samples and training the generator model to predict the missing taxa based on the remainder of the sample. For the second step, we use the trained generator to produce perturbed microbiome samples by replacing all the masked taxa with generator predictions and train a discriminator model to differentiate the original taxa of the sample from those replaced by the generator. Essentially, the generator attempts to fill in the masks with taxa predictions and the discriminator takes in the predicted sequence and attempts to identify which taxa are modified by the generator.

Both the generator and discriminator models have the same general architecture as shown in Fig 3. To train the generator, the inputs are randomly corrupted by replacing 15% of taxa IDs with a special 'mask' ID, and the embedding of each masked taxa after the final encoder layer is fed into a classification head to predict the ID of the masked taxa.

To train the discriminator model, we take the masked sample completed by the generator as input to the discriminator, and feed the embedding of each taxon after the final encoder layer into a classification head that differentiates 'real' (original taxa) from 'modified' (generated taxa).

At the end of pre-training, we have two transformer models, the generator and discriminator. Following the practice of the original work, we use the encoder of the pre-trained discriminator as the initial model to be fine tuned for downstream tasks.

*Pre-training details.*

We perform pre-training on 18,480 gut microbiome samples from the American Gut Project Database using the ELECTRA scheme as described above. Specifically, the generator was trained for 240 epochs to predict masked microbe embeddings, and checkpoints of the model

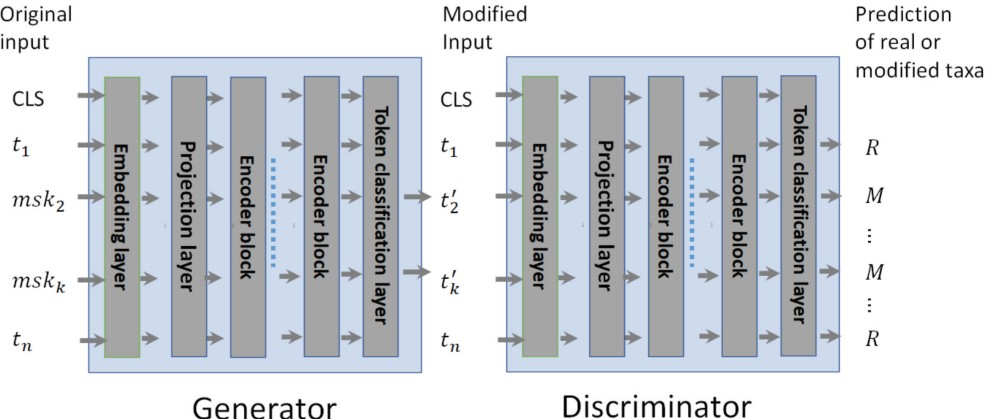

**Fig 3. Electra pre-training diagram.** A generator is trained to predict the masked taxa from a sample. A discriminator is trained to differentiate taxa filled in by the generator from the original taxa in the sample. Both use the same transformer architecture, and have token level classification heads. The generator token level classification head predicts the taxa ID whereas the discriminator token level classification head predicts the input taxa as "Real" or "Modified".

were saved every 30 epochs. The discriminator was then trained on the replacement prediction task for 120 epochs with replacements generated by the increasingly trained generator. Specifically, for every 15 epochs of discriminator training, we replace the generator used to produce inputs for the discriminator with a stronger generator using the previously mentioned checkpoints. For example, the generator trained for 30 epochs provided inputs for the first 15 epochs of discriminator training. Then, for epochs 16-30, discriminator inputs were provided by the generator trained for 60 epochs. This was done to gradually ramp up the difficulty of the replacement prediction task.

**Architecture and pre-training choices.** We performed model architecture selection on the basis of pretraining results. We used 16,000 AGP samples to perform the training for both the generator and discriminator models, and used the remaining 2,480 samples as a hold-out validation set to decide the model architecture as well as the stopping point for the pretraining. Specifically, we observed that fewer than 5 layers of encoders leads to reduced capacity for the discriminator to differentiate between real and imputed taxa, whereas a larger number of layers does not produce noticeable benefit. We additionally chose to stop the discriminator's pretraining at 120 epochs because we observed its prediction accuracy on the holdout set stabilizing at that point, even when substituting in better-trained generators.

**Task specific fine-tuning.** Given a specific prediction task and the pre-trained discriminator, we remove the token classification head and add a new (randomly initialized) sequence classification head to the 'CLS' token. In addition to the embedding layer, we also freeze the parameters of the encoder blocks such that only the classification head and the projection layer were trained during fine-tuning. In other words, the pre-trained discriminator encoders are used as a universal encoder for representing microbiome samples for different prediction tasks. Empirically we have found this practice reduces overfitting and produces more robust generalization performance across different tasks.

*Fine-tuning details.*

We perform fine-tuning using Stochastic Gradient Descent (SGD) optimization with a learning rate of 0.01, momentum of 0.9, and the mean squared error loss, which we found gave better results than the more traditional negative cross-entropy loss, potentially because mean squared error is more robust to noise and outliers. Furthermore, during training, we perform data augmentation by randomly deleting 10% of the input taxa (meaning we randomly select one in ten of the taxa in the data point and remove them from the input sequence, similar to the method introduced by [27]) in each training sample to increase the robustness of the trained model and reduce overfitting. The SGD optimization is performed for a total of 50 epochs on the training subsets of the labeled AGP data. As the labeled AGP datasets have highly unbalanced labels (Table 1), we oversample the minority class to ensure the model sees equal numbers of samples from each class. We use cross-validation on a subset of the IBD data to tune the hyperparameters (random deletion percentage for data augmentation and the choice of MSE vs the Cross Entropy loss) for fine-tuning.

**Table 1. Three classification tasks derived from the AGP data and meta data.**

| Datasets | AGP | AGP-IBD | AGP-Fruit | AGP-Vegetable |
|---|---|---|---|---|
| # of samples | 18480 | 8571 | 6540 | 6549 |
| # of positive samples | N/A | 435 | 4026 | 5654 |
| # of negative samples | N/A | 8136 | 1514 | 895 |

## 2.3 Feature ablation attribution: finding the important taxa

We are interested in finding which microbial taxa the model relies on most for making a positive or negative classification of the samples. To this end, we use feature ablation attribution [28].

Consider a sample $X$ containing $n$ microbial taxa, which the model predicts as being positive (for some property, e.g, IBD) with probability $M(X)$. Feature ablation individually deletes each microbe taxon from the original $X$, then records how much each taxon's removal reduces the model's predicted probability of being positive. We average these changes across every sample in which a taxon appears, giving the expected change in classification probability caused by deleting the taxon in question from a random sample containing the taxon.

Given a dataset $\mathbf{D}$, let $\mathbf{D}_m$ denote the set of samples that contains a specific microbe $m$, we can calculate $m$'s attribution $a(m)$ as:

$$a(m) = \frac{1}{|\mathbf{D}_m|} \sum_{X \in \mathbf{D}_m} \mathbf{M}(X) - \mathbf{M}(X \setminus m) \tag{1}$$

where $\mathbf{M}(\cdot)$ denotes the model's probabilistic output for the given input and $X \setminus m$ denotes sample $X$ with microbe $m$ removed.

## 2.4 Datasets

We use three different datasets over the course of this study. We now describe them and summarize where they are used.

*American Gut Project (AGP).*

The American Gut Project (AGP)[8] is a crowdsourced microbiome data gathering effort. From it, we used 18,480 microbiome samples sequenced from the v4 hypervariable region of the 16S gene that were curated by the authors of [13]. The sample sequences come with metadata information on the subject the sample originates from, providing information about their diet, medical status on inflammatory bowel disease and more. We used all 18480 samples for our pre-training and relevant portions in our evaluation of downstream tasks. We now describe the three downstream tasks we ran experiments on.

- Inflammatory Bowel Disease (IBD). This task aims to predict whether a given microbiome sample belongs to an individual diagnosed with IBD or not. Samples originating from individuals with IBD are the positive class. Label information was drawn from AGP metadata producing 435 samples from IBD positive individuals and 8,136 healthy controls.
- Frequency of fruit in diet. This task aims to determine the frequency with which an individual consumes fruits based on their microbiome sample. The label is derived from AGP metadata, which ranks fruit consumption frequency on a one to five scale. For this experiment, samples ranked 3-5 are grouped to form the positive (frequent) class. Samples ranked 0-2 are considered negative (infrequent). Out of 6,540 AGP examples with fruit metadata, 4,026 were labeled positive.
- Frequency of vegetable in diet. This task aims to determine the frequency with which an individual consumes vegetables based on their microbiome sample. In the same manner as the fruit task, label information was drawn from the AGP metadata and frequency ranks from 0–5 were grouped to form the "frequent" (3–5) and "infrequent" (0–2) classes. Out of 6,549 AGP examples containing vegetable frequency metadata, 5654 were labeled positive.

**Table 2. Runtimes and estimate costs of different experiments performed in this paper.** All runtimes were measured on a single Nvidia A40 GPU, and costs are estimated based on the hourly price of $0.403 required to rent an Nvidia A40 from vast.ai as of 03/11/2024.

| Experiment name | Time (hr) | Cost ($) |
| --- | --- | --- |
| Pretraining | 23.43 | 9.44 |
| Fine-tuning IBD (5 runs) | 12.20 | 4.92 |
| Fine-tuning Fruit (5 runs) | 13.98 | 5.63 |
| Fine-tuning Vegetable (5 runs) | 10.74 | 4.33 |

Table 1 provides the summary statistics for the three classification tasks. Table 2 provides the run times and costs required to perform the pretraining and 5 training runs on the relevant portions of AGP.

*Halfvarson (HV).*

This dataset comes from an IBD study performed in [29]. We used the curated dataset produced in [13], which contains 564 microbiome samples, with 510 of them IBD positive.

*HMP2.*

This dataset comes from an IBD study performed as part of phase 2 of the Human Microbiome Project [30]. Again, we used the curated dataset produced in [13], which contains 197 microbiome samples with 155 IBD positive examples.

Because AGP-IBD, AGP-Fruit and AGP-Vegetable all derive from the larger AGP dataset, there is overlap between the data used for model development and the evaluation data that provide the results in Table 3. Specifically, both the GLoVE embeddings from [13] and our own pretrained model are trained on the full 18,480 sample AGP dataset. However, neither process has access to any of the *labels* for AGP-IBD, AGP-Fruit or AGP-Vegetable, only the unlabeled taxa sequences associated with the samples. Additionally, each dataset includes at least some patients from which multiple samples were taken. When both training and testing on AGP (as in Table 3), we employ patient–level blocking of data between training, validation, and testing sets. We ensure a fair comparison between our approach and the baselines by providing all baselines with equivalent access to both unsupervised and labeled data across every evaluation. Thus, any baseline with a representation learning phase will use the same 18,480 AGP samples as our method.

## 3 Results and discussions

### 3.1 Transformer representations outperform representation learning baselines on multiple microbiome tasks

In this section, we empirically compare transformer-produced sample representations against a variety of baseline methods. Our baselines include **Weighted**, a simple non-contextualized abundance-weighted-averaging of the GloVe embeddings from [13], two classic dimension reduction based methods, and two deep learning based methods introduced by [32], each of which performs dimension reduction using the sample taxonomic abundance profiles as input features. Additionally, we include a classical random forest based classifier that directly uses the sample taxonomic abundance profile as features with no dimension reduction:

- **No dimension reduction (No-DR)**: This baseline skips any representation learning and directly trains a random forest model on the taxonomic abundance tables, representing each sample as a 26,726 dimensional vector of ASV abundances.

- **PCA**: Principle Component Analysis, configured to retain at least 99% of the variance.
- **RandP**: Random Gaussian Projection, relying on the Johnson-Lindenstrauss lemma [33] and implemented with scikit-learn [34] using eps 0.5.
- **AE**: An MLP-based autoencoder architecture [35], with two sizes: $AE_{Best}$ (28.4M parameters) and $AE_{Match}$ (7.2M parameters).
- **CAE**: An convolutional neural network-based autoencoder architecture [36], with two sizes: $CAE_{Best}$ (12.3K parameters) and $CAE_{Match}$ (102.6K parameters).
- **Weighted**: The abundance-weighted average of a sample's GloVe representations.

We used a reduced training set to quickly sweep the full range of model hyperparameters described in [32] for their effectiveness in our setting. We found that the variational autoencoder failed to produce useful results, regardless of hyperparameters, and thus omitted this architecture in the comparisons. For the two remaining architecture (AE and CAE), we selected two sizes: one that achieved the best validation performance using the reduced training set ($CAE_{Best}$), and another that aims to match the parameter count of our own model (7.07M) as closely as possible.

For the baselines from [32], we adapt that work's random forest classification layer (and the range of hyperparameters to consider), because random forest most consistently achieved the best performance across the settings [32] explored.

As mentioned previously, our method applies a standard multi-layered perceptron (MLP) classifier to the transformer-produced sample representations for classification. To allow Weighted to act as a more consistent comparison with our model, we replaced the random forest classifier used in prior work with the same MLP classifier. We evaluate our method and the baseline methods using the AGP dataset on three microbiome classification tasks.

For each method and task, we perform 5 training runs. Our methods (meaning the Transformer and Weighted baseline) adopt the evaluation framework described in [32] to decide the stopping epoch: each run first blocks out 20% of the data to be used only for testing, then splits the remaining 80% into train and validation subsets to decide the best stopping epoch. Then, the 80% of non-test data is recombined into a single training set, and the model is re-finetuned from scratch on the non-test data using the discovered stopping epoch. Note that PCA, RandP, AE, and CAE baselines also use the train/validation split of non-test data from [32] to tune the random forest hyperparameters in addition to stopping epoch.

We consider two different evaluation criteria: the Area Under the ROC Curve (AUROC) and the Area Under the Precision-Recall curve (AUPR). We select these two metrics because they allow us to rigorously compare the discriminative capabilities of our models and baselines on unbalanced classes, without having to specify a particular threshold for what we consider a "positive" or "negative" classification.

Table 3 shows the performance of all methods on three tasks. Among representation learning based methods, we see that for the IBD and Fruit tasks, the transformer produced representation achieved substantially improved performance for both AUROC and AUPR. Performances on the Vegetable task are much closer together across methods, especially between Weighted, PCA and Transformer, with PCA even marginally edging out Transformer's AUPR score. This confirms that our approach learns a transformer model that produces robust sample representation that performs well across multiple prediction tasks.

The simple, random forest based classifier shows the best performance across the board. We include this method as a point of comparison, reflecting the fact that well-executed classical approaches often beat more complex representation learning methods [37–39]. However, our aim is *not* to simply learn the most predictively accurate microbiome phenotype classification model. Rather, we aim to capture generalizable patterns of microbiome community

**Table 3. Average performance (standard deviation) on three tasks within the AGP dataset. The best performance achieved with representation learning, along with entries whose difference from it is not statistically significant at $p < 0.05$, are bolded. No-DR's performance is shown as a separate category as it does not involve representation learning, and is marked with an asterisk ("*") when it is statistically significantly better.**

| | IBD Task | | Fruit Diet Task | | Vegetable Diet Task | |
|---|---|---|---|---|---|---|
| | AUROC | AUPR | AUROC | AUPR | AUROC | AUPR |
| Transformer | **0.687**±.04 | **0.121**±.02 | **0.619**±.02 | **0.707**±.02 | **0.700**±.02 | **0.928**±.01 |
| Weighted | 0.646±.02 | 0.089±.02 | 0.585±.02 | **0.674**±.04 | **0.695**±.02 | **0.930**±.01 |
| PCA | 0.571±.04 | 0.082±.02 | 0.576±.03 | **0.689**±.04 | **0.700**±.01 | **0.932**±.01 |
| RandP | 0.621±.03 | 0.095±.02 | 0.540±.03 | 0.653±.04 | 0.669±.02 | **0.926**±.01 |
| $AE_{Best}$ | 0.576±.05 | 0.090±.02 | 0.532±.03 | 0.647±.05 | 0.654±.02 | **0.922**±.01 |
| $AE_{Match}$ | 0.604±.06 | **0.097**±.03 | 0.542±.01 | 0.660±.04 | 0.669±.02 | **0.926**±.01 |
| $CAE_{Best}$ | 0.625±.03 | **0.093**±.03 | 0.571±.03 | **0.677**±.05 | **0.662**±.06 | **0.920**±.03 |
| $CAE_{Match}$ | 0.607±.03 | 0.086±.02 | 0.563±.02 | **0.675**±.04 | **0.684**±.02 | **0.927**±.01 |
| No-DR | 0.717±.05 | 0.151±.05 | 0.655*±.03 | 0.749*±.04 | 0.729*±.01 | 0.939±.01 |

structure and function. As we will see, our learned contextualized representations allow for unprecedented generalization to novel datasets and reflect meaningful biological structure.

## 3.2 Generalization to independent datasets

One of the largest challenges in working with microbiome data is that there is large variance in the distributions and characteristics of data used from study to study. Therefore it is important to test how well our transformer based prediction models generalize on independent datasets that come from different population/sample distributions. To test this, we applied our transformer model trained for the IBD prediction task using the AGP data on the Halfvarson and HMP2 datasets from independent studies, without finetuning our model on any data from those independent studies.

An issue that arises when performing such cross-study tests is the need to decide a stopping point during finetuning to pick the best model to use on the test data. In the previous single study experiments, using a held-out validation set for this purpose proved to be an effective strategy. However, due to the substantial distributional shift between the AGP data used for training/validation and the independent test set of Halfvarson and HMP2, using a held-out AGP validation set for stopping is observed to lead to poor and highly unstable results (shown by "Transformer (original)" in Table 4). We address this problem by introducing a simple ensemble strategy. During fine tuning, we train an ensemble of $k$ classifiers using different random initializations of the classification head. Similar to the standard practice when applying transformer to language [26], we found that each individual classifier only needs to be fine-tuned for a single epoch, i.e., going over all of the training once, and that training more epochs often leads to overfitting. In our experiments, we used ensemble size $k = 10$.

We compare our ensemble performance with the baselines described above, and additionally strengthen the Weighted baseline of [13] by using an ensembled MLP classifier and reporting the best *testing* performance achieved by the Weighted baseline method during training. The baselines from [32] use random forest as the classifier and do not have a similar free parameter regarding their stopping condition.

We report the performance of all methods averaged across five random runs with different initialization in Table 4. The results show that our method consistently achieves better performance on the Halfvarson dataset compared to all baselines, and comparable performance on the HMP2 dataset compared to the best performing of the Weighted baseline model selected

**Table 4. Average performance (standard deviation) on independent IBD datasets. Best performer and entries whose difference from it is not statistically significant at $p < 0.05$ are bolded.**

|  | HMP2 | | Halfvarson | |
|---|---|---|---|---|
|  | AUC | AUPR | AUC | AUPR |
| Transformer (ensemble) | **0.682**±.02 | **0.855**±.01 | **0.805**±.01 | **0.973**±.00 |
| Transformer (original) | 0.460±.03 | 0.773±.02 | 0.719±.09 | **0.957**±.02 |
| Weighted (ensemble) | 0.668±.00 | **0.863**±.00 | 0.752±.00 | 0.962±.00 |
| PCA | 0.570±.02 | 0.795±.01 | 0.578±.06 | 0.931±.01 |
| RandP | 0.583±.03 | 0.813±.02 | 0.509±.03 | 0.909±.01 |
| AE$_{Best}$ | 0.618±.02 | 0.839±.01 | 0.519±.02 | 0.912±.01 |
| AE$_{Match}$ | 0.644±.02 | **0.850**±.01 | 0.499±.05 | 0.903±.02 |
| CAE$_{Best}$ | **0.697**±.01 | **0.879**±.01 | 0.426±.04 | 0.890±.01 |
| CAE$_{Match}$ | **0.706**±.04 | **0.883**±.04 | 0.488±.04 | 0.906±.01 |
| No-DR | 0.657±.01 | **0.869**±.01 | 0.530±.01 | 0.926±.00 |

using testing data. Although CAE$_{Best}$ and CAE$_{Match}$ achieve slightly higher HMP2 performance these differences are not statistically significant and come at the cost of an enormous deficit on Halfvarson. No-DR demonstrates competitive performance for HMP2 in terms of the AUPR score but performs significantly worse in the AUC score. For Halfvarson, No-DR's performance drops substantially on both measures. Notably, HMP2 is more similar to the AGP data, with 51% of its ASVs overlapping with AGP (measured using similarity threshold of 99% and e-value threshold of $10^{-20}$), whereas only 34% of Halfvarson's ASVs overlap with AGP. The dominant performance of our method highlights our approach's ability to generalize well to out-of-distribution settings, particularly when faced with large distribution shifts.

## 3.3 Context sensitive taxa embedding captures biologically meaningful information

We hypothesize that the superior generalization of our model is because our pre-trained language model transforms the input taxa embedding into a more meaningful latent space capturing context sensitive information (Fig 1G), making biologically relevant features of the taxa more readily extracted and applied to downstream tasks.

*Taxonomic association.*

We focus on the top 5,000 (out of 26,726) most frequent taxa from the IBD dataset and compute their averaged contextualized embeddings across every entry in the IBD dataset. Fig 4 shows the t-SNE [40] visualization of the taxa using the original vocabulary embedding from [13] (Fig 4A) and the averaged contextualized embeddings produced by our model (Fig 4B), colored by the phylum of the taxa assigned by the DADA2 tool [41]. t-SNE is better suited to capturing the local neighborhood than the global structure, with points close together in the t-SNE visualization also generally being close together in the original embedding space. However, t-SNE gives a much worse impression of the overall (global) shape of the data [42].

From Fig 4, we see that the original embedding space in (Fig 4A) displays a degree of clustering by phylum. In particular, Proteobacteria (red) tend to cluster in distinct manifolds from the rest of the taxa. However, most of the taxa lie in a single large but stratified manifold of mixed phyla. In contrast, the contextualized representations in Fig 4B appear to have more consistent clustering by phylum in this reduced 2-D space. Given the potential for 2-D projection methods, such as t-SNE, to introduce visualization artifacts, we further compare the

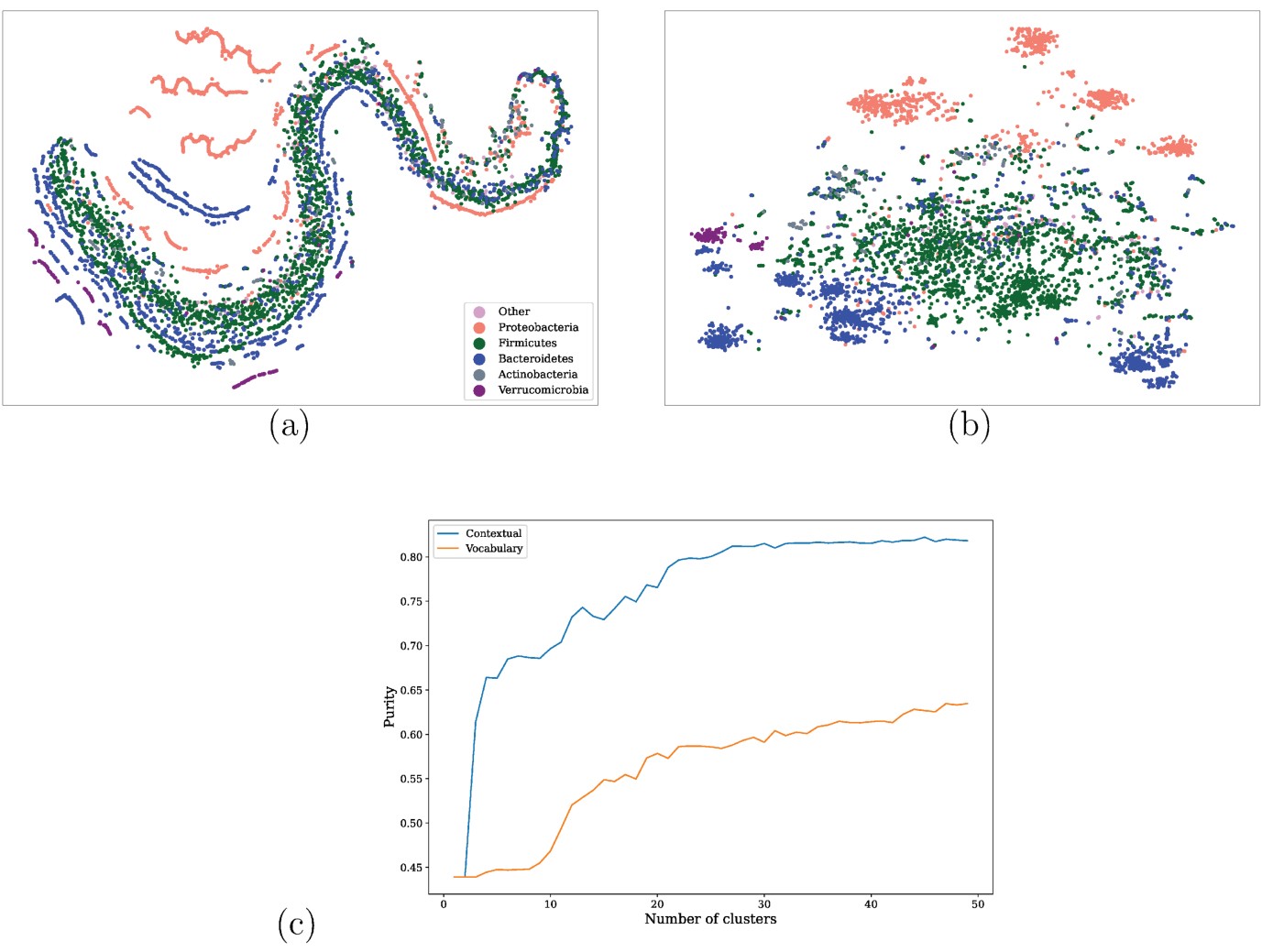

**Fig 4. Fig 4A shows t-SNE visualization of original taxa vocabulary embeddings, and Fig 4B visualizes the contextualize taxa embeddings.** Both are colored by phylum. See Fig S1 for embedding spaces colored by phylum, class, order, and family. Fig 4C shows the phylum purity of clusters versus K for K-means clustering in the vocabulary and contextualized embedding spaces, showing the tighter clustering of the embedding space isn't simply an artifact of the t-SNE dimension reduction.

taxonomic associations with each embedding spaces by performing $k$-means clustering separately on each full-dimensional embedding space and comparing the phylum purity curves as the number of clusters ($k$) varies. Fig 4C shows that clusters formed in the contextualized embedding space have consistently and substantially less cross-phylum contamination (as shown by higher phylum purity) as compared to clusters in the GloVe embedding space. The difference in clustering purity is remarkable, with clusters in the contextualize embedding approximately 20 percentage points more pure than clusters in GloVe embeddings.

To highlight the differences between the two representations, Fig 5 explores the mapping between them by highlighting the same group of taxa in both figures, where the left column shows the t-SNE visualization of the original taxa embeddings, and the right column shows the t-SNE of the contextualized taxa embeddings. From the comparison, we can see that the phyla that are well separated in the original embedding space as distinct manifolds are well preserved and further compacted into tighter clusters (see Fig 5A).

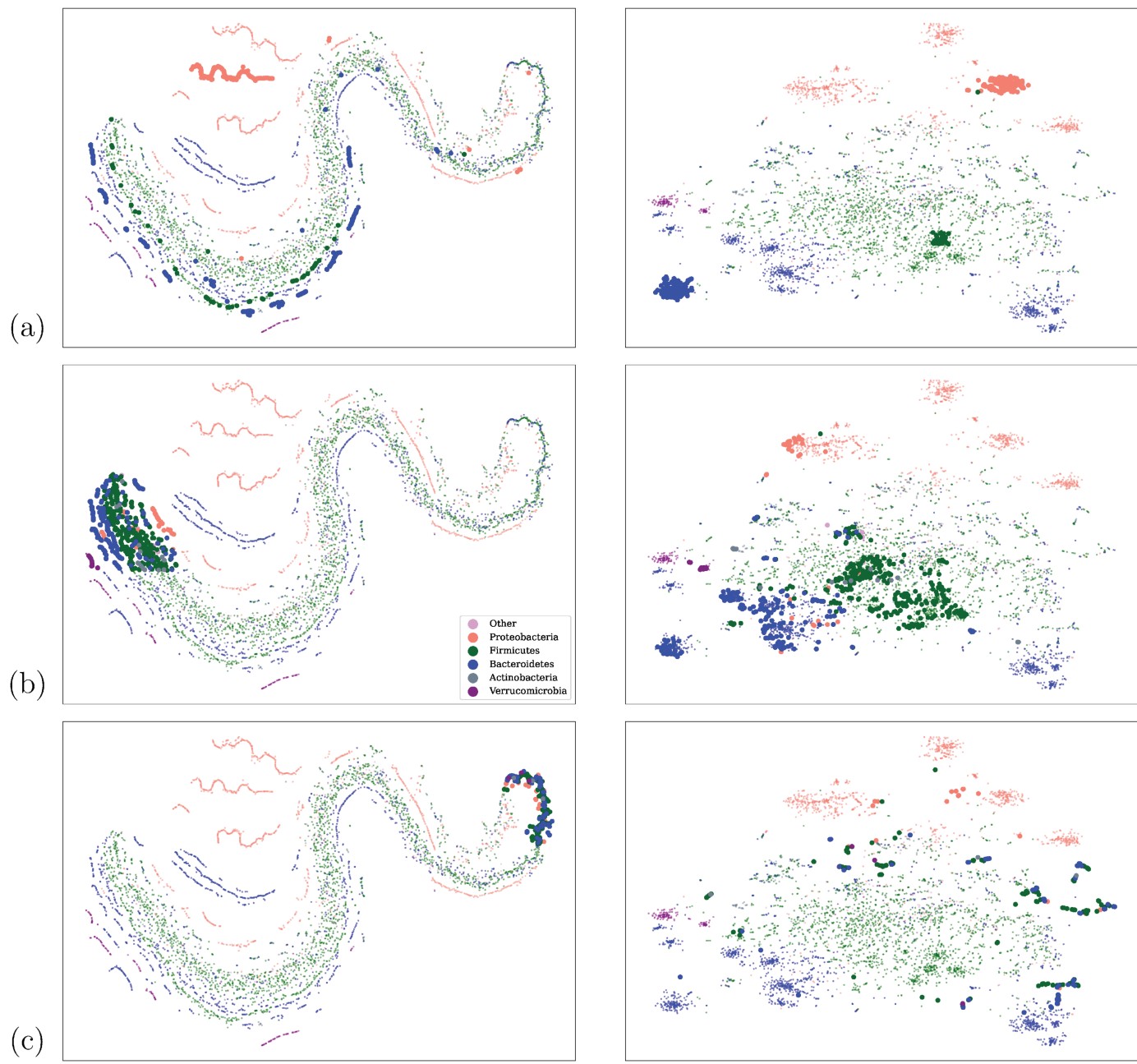

**Fig 5. Mapping between the original vocabulary and contextualized embedding spaces.** Fig 5A shows how the contextualized embeddings can extract "threads" of a single phylum from the vocabulary embedding space, and map those taxa to tight clusters in the contextualized embeddings. Fig 5B shows that the mapping to the contextual embedding space is able to more cleanly separate taxa by phylum. Fig 5C contrasts Fig 5B and shows that taxa which are very tightly clustered in the vocabulary embeddings may not map to meaningful clusters or phylum-level separation in the contextualized embedding space.

The data in the original vocabulary embedding space appear to lie on long "strands", rather than clump together in clusters. In particular, we see a large strand in the middle that contains most of the data, and seems to be made up of smaller "threads" very close together. The contextual representations appear to "unwind" the large strand so that the smaller threads can be extracted and grouped together in their own isolated clusters, which more cleanly separate by

phylum (see Fig 5A). This highlights the capability of self-supervised representation learning to flexibly extract important features from unlabeled data.

Our model's ability to cluster taxa by phylum seems to degrade for taxa whose vocabulary embeddings are too close together. Fig 5B highlights taxa in a less compact region of the original embedding space, and highlights the same taxa in the contextualized embedding space, where the taxa show reasonable separation by phylum. In comparison, Fig 5C highlights a more compact region of the vocabulary space, as well as the corresponding taxa in the contextualized embedding space, which appear to show worse separation than we see in Fig 5B.

*Correlation with Metabolic pathways.*

Similar to [13], we investigate whether our contextualized embedding dimensions correlate with known metabolic pathways. We map the vocabulary taxa ASVs to their nearest neighbors in the KEGG database [43] using Piphillin [44], following the method used by [13]. Metabolic pathways for each mapped ASV are then extracted using the KEGGREST API [45], leading to a total of 141 pathways. Each ASV is represented using a one-hot encoding of the 141 metabolic pathways, assigning a 0 if the ASV is not involved in the pathway, and a 1 if it is involved.

We limit the following analysis to ASVs involved in at least one of the 141 pathways, resulting in 11,893 ASVs, each represented by a 141-dimension binary vector indicating their involvement in the extracted pathways. We have seven fewer pathways than were present in the metabolic pathways analysis of [13], due to changes in the KEGG [43] database.

We compute the Spearman's correlation between each of our contextualized embedding dimensions and the 141 extracted metabolic pathways, producing a 200 by 141 correlation matrix. The same process is repeated for the 100-dimensional GloVe embedding, producing a 100 by 141 correlation matrix. Fig 6 shows both sets of correlations using heatmaps. We can see that, although both embeddings show clear correlations with some metabolic pathways, the contextualized embedding dimensions capture stronger correlation, signified by the darker blue and red colors in the heatmap. To assess the statistical significance of the observed correlations, we applied a permutation test with 1,000 permutations. This test generates a distribution of correlations under the null hypothesis that the embeddings and the pathways are independent. We consider a correlation statistically significant if its value exceeds all the correlations observed in the simulated null distribution (effectively corresponding to a p-value of less than 0.001) and include only those correlations that are significant. We then compare the strengths of the remaining statistically significant correlations found for our contextualized embeddings to those found for the GloVe embeddings, by contrasting the distribution of the filtered correlation magnitudes from both embeddings in Fig 7, which visually shows that the normalized histograms of the contextualized embedding dimensions are shifted to the right compared to that of the GloVe embedding dimensions.

To verify that the two distributions of correlation magnitude are indeed different, we perform two different non-parametric statistical tests: the Kolmogorov–Smirnov two-sample test [46,47] and the Epps–Singleton two-sample test [48] using SciPy's [49] implementation. Both tests reject the null hypothesis that the two distributions are equivalent with p-values of $4.19 \times 10^{-26}$ and $9.71 \times 10^{-50}$, respectively. The effect size, measured by Cliff's delta [50], is 0.2.

## 3.4 Understanding taxa importance for IBD prediction

In this part, we focus on the fine-tuned IBD ensemble prediction model to understand what taxa play critical roles in our model's IBD prediction by studying their attribution. We first

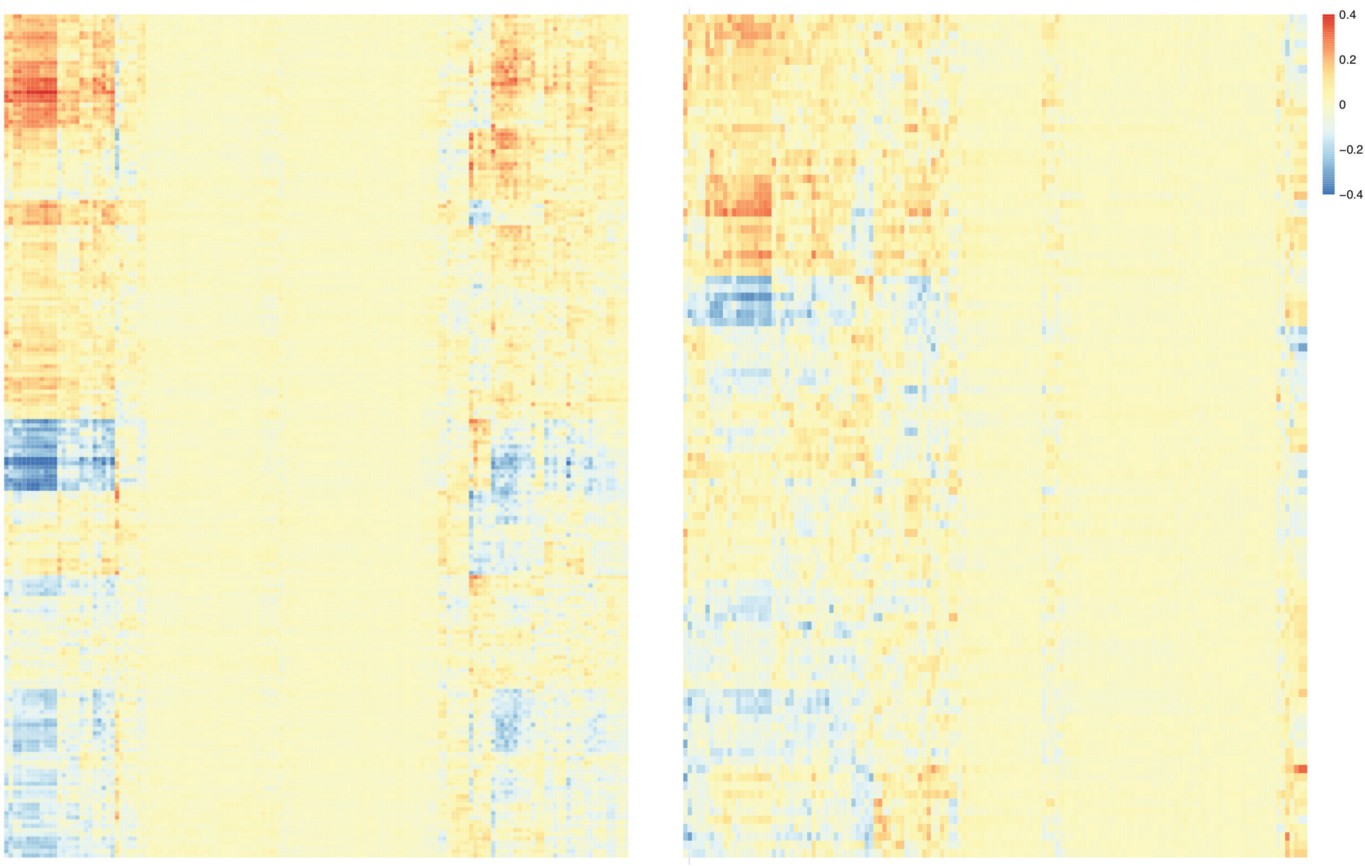

**Fig 6.  Heatmaps showing how strongly each embedding dimension (y-axis) correlates with each metabolic pathway (x-axis), for both our contextualized embeddings and the prior GloVe embeddings.**

consider the 5,000 most frequent taxa shown in Fig 4 and compute for each taxon its average attribution toward the model's IBD prediction using the AGP IBD data, as described in Sect 2.3.

Fig 8A presents the t-SNE visualizations of the contextualized embeddings colored by taxa attribution strength. The visualization shows multiple clusters of high and low attribution taxa, indicating that local neighborhood distances in the original embedding space reflect taxa attributions. It is important to note that the contextualized embeddings generated by our pre-trained language model have never been trained on any IBD labels, yet their local structure appears to reflect taxa attributions, suggesting that our pre-trained language model indeed captures meaningful biological information.

Next, we wish to find the most important taxa for our model's correct IBD classifications across different study populations. We therefore filter the data to focus on samples for which our model makes confident and correct predictions. Specifically, we filter each of the three IBD datasets (American Gut Project (AGP) [8], the curated [13] versions of the Human Microbiome Project phase 2, (HMP2) [30], and Halfvarson (HV) [29]) and include only correctly classified samples with a predicted probability ranking within the top 50%, regardless of being positive or negative. To focus on reasonably common microbial taxa, we also filter

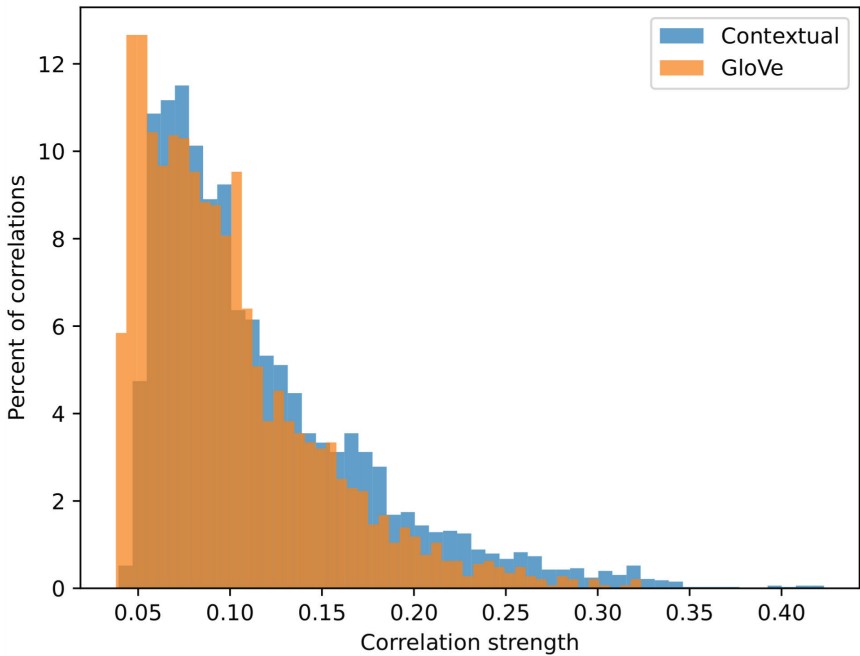

**Fig 7.   Distribution of the magnitude of statistically significant correlations between embedding dimensions and metabolic pathways, for both contextualized embeddings and the prior GloVe embeddings.**

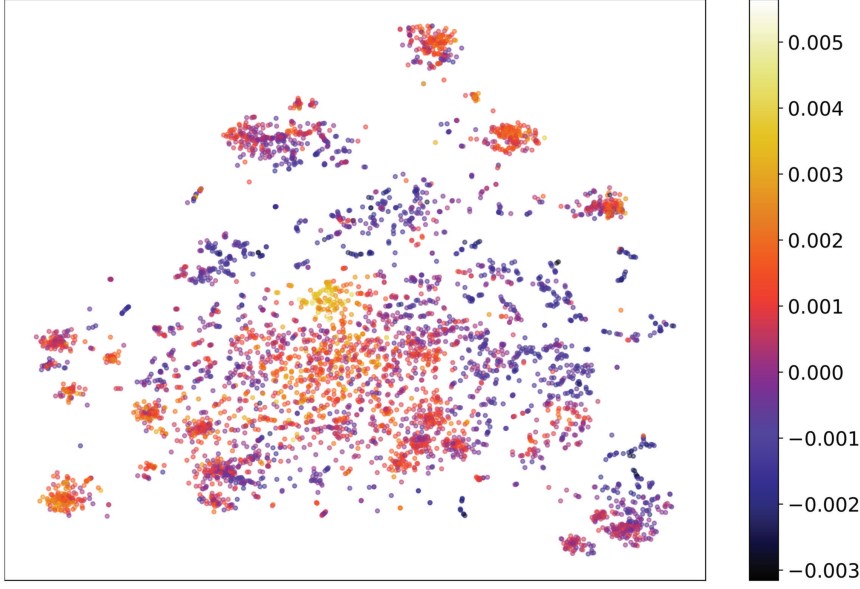

**Fig 8.   t-SNE visualization of the contextualized embeddings colored by attribution to IBD.** The taxa associated with IBD are visualized in lighter color (yellow) and the taxa associated with no-disease state are in dark purple.

out taxa that appear in less than 5% of all samples across all three IBD datasets (AGP, HMP2 and HV).

To allow for independent validation of our attribution estimation, we combine HV and HMP2, into a single dataset (HV+HMP2), filter for correct confidence again, and compute the average attribution on APG and HV+HMP2 separately, and reduce noise by filtering out any taxa that appear in less than five samples in each dataset. The attribution for a taxon is considered validated if it has two estimates from AGP and HV+HMP2 respectively, and they have the same sign. Of the 5,716 taxa that appear in HV+HMP2, 695 appear in at least 5% of the combined IBD-labeled data points, 530 of those appear in at least five confident and correct samples in HV+HMP2 (and have been assigned attributions), and 399 of those taxa have matching signs in their attributions between the AGP data and the HV+HMP2 data. This ensures that we only identify these microbial taxa that have consistent impact on the model in two different populations. We then compute the average attribution of all validated taxa across the combined (filtered) datasets. We show the 10 taxa most attributed to negative IBD classification in Table 5 and the 10 taxa most attributed to positive IBD classification in Table 6. Sequences for these taxa are available in S1 and S2 Tables, respectively.

**Comparing identified taxa to existing data repository.** We compared the top 10 ASV attributions to IBD and the healthy cohort (20 ASVs total) found with our model to 284 markers taxa identified in the data repository for the human gut microbiota [51] across three projects (NCBI PRJEB7949 (95 entries), NCBI PRJNA368966 (32 entries), NCBI PRJNA3x85949 (157 entries)) comparing IBD and healthy controls (query request: gmrepo.humangut.info/phenotypes/comparisons/D006262/D015212).

Due to the difference in technologies between all the datasets, we compare the markers across the studies at the genus level. In our study, seven ASVs were not resolved beyond the

**Table 5. Top 10 Taxa associated with negative (non-disease) IBD classification ordered by attribution strength.**

| Phylum | Class | Order | Family | Genus |
|---|---|---|---|---|
| Firmicutes | Clostridia | Clostridiales | Lachnospiraceae | NA |
| Proteobacteria | Gammaproteobacteria | Betaproteobacteriales | Burkholderiaceae | Sutterella |
| Bacteroidetes | Bacteroidia | Bacteroidales | Prevotellaceae | Prevotella |
| Proteobacteria | Gammaproteobacteria | Pasteurellales | Pasteurellaceae | NA |
| Bacteroidetes | Bacteroidia | Bacteroidales | Prevotellaceae | Prevotella_9 |
| Fusobacteria | Fusobacteriia | Fusobacteriales | Fusobacteriaceae | Fusobacterium |
| Firmicutes | Bacilli | Lactobacillales | Lactobacillaceae | Lactobacillus |
| Bacteroidetes | Bacteroidia | Bacteroidales | Muribaculaceae | NA |
| Bacteroidetes | Bacteroidia | Bacteroidales | Bacteroidaceae | Bacteroides |
| Firmicutes | Clostridia | Clostridiales | Lachnospiraceae | NA |

**Table 6. Top ten Taxa associated with positive IBD classification ordered by attribution strength.**

| Phylum | Class | Order | Family | Genus |
|---|---|---|---|---|
| Proteobacteria | Gammaproteobacteria | Betaproteobacteriales | Burkholderiaceae | Sutterella |
| Firmicutes | Clostridia | Clostridiales | Ruminococcaceae | Ruminiclostridium_5 |
| Bacteroidetes | Bacteroidia | Bacteroidales | Tannerellaceae | Parabacteroides |
| Firmicutes | Clostridia | Clostridiales | Lachnospiraceae | NA |
| Firmicutes | Clostridia | Clostridiales | Ruminococcaceae | Ruminococcaceae_UCG-9 |
| Firmicutes | Clostridia | Clostridiales | NA | NA |
| Firmicutes | Clostridia | Clostridiales | Ruminococcaceae | Candidatus_Soleaferrea |
| Firmicutes | Clostridia | Clostridiales | Lachnospiraceae | Lachnoclostridium |
| Firmicutes | NA | NA | NA | NA |
| Firmicutes | Clostridia | Clostridiales | Lachnospiraceae | Anaerostipes |

family level, and are therefore excluded from this analysis. Further, two of our ASVs belonged to sub-clade of a genus, we considered them belonging to the genus of the clade: specifically Prevotella_9 (which was considered Prevotella in this analysis) and Ruminoccocus_1 (which was considered Ruminoccocus in this analysis).

Out of our 13 ASVs, four ASVs belong to genera Prevotella, Paraprevotella, and Lachnoclostridium, which were also found to be consistently associated with the healthy cohort in the data repository for the human gut microbiota (DRHM). Therefore they constitute consistent markers with the previous literature. One ASV, belonging to the genus Atopobium, was only associated with IBD in both our study and the DRHM, also constituting a markers of IBD consistent across our study and the database. Out of the remaining eight, we found two new ASVs markers that were not previously identified: the genera Allisonella (Associated with health) and Methanosphaera (associated with IBD).

Finally, the six remaining ASVs showed mixed patterns in the DRHM, where some taxa of the genus seem to be a marker for the IBD and other taxa are enriched in healthy individuals. Out of these six genera, three markers mostly agree with our results: Bacteroides, which was associated with healthy individuals in 17/20 taxa, Ruminococcus showing the same pattern in 5/9 taxa, and Rosebduria also with the same pattern for 2/3 taxa. The other three genera show the opposite trend when comparing the DRHM markers with our work. Most notably, Lactobacillus is associated with the healthy cohort in our analysis, while 8/9 markers from this genus are enriched in the IBD cohort in the DRHM. We see more nuanced results for the genera Parabacteroides where 3/7 markers are associated with the control cohort in the DRHM (and a marker of IBD for us), as well as Oscillibacter, associated with the healthy individuals in 2/3 taxa in the database, which contradict our finding.

In summary, out of the 13 ASVs resolved at the genus level from our study, our analysis revealed two new markers not included in the DRHM. For five of these ASVs, our result is consistent with the DRHM markers. For the remaining three, we see mixed results. Here, the taxonomic resolution of our 16S becomes a limiting factor as these genera show different behavior at the species level.

## 4 Conclusion and future work

We apply recent natural language processing techniques to learn a language model for microbiomes from public domain human gut microbiome data. The pre-trained language model provides powerful contextualized representations of microbial communities and can be broadly applied as a starting point for any downstream prediction tasks involving human gut microbiome. In this work, we show the power of the pre-trained model by fine-tuning the representations for IBD disease state and diet classification tasks, achieving strong performance in all tasks. For IBD, our learned representations enable an ensemble model with competitive performance that is robust across study populations even with strong distributional shifts.

We visualize the contextualized taxa embedding learned by our pre-trained language model and show that it captures biologically meaningful information including phylogenetic structure, previously discovered metabolic pathways and IBD association without any prior training on such signals. We employ an interpretability technique to investigate the basis for our models' IBD classification decisions and identify sets of taxa that negatively and positively attribute to the model's predictions. We find known biomarkers of both IBD and gut homeostasis, as well as evidence that our embeddings learn to separate ASVs by their pathogenicity, even among ASVs sharing the same family and genus level phylogenetic classifications.

Our investigation suggests that NLP techniques like deep language models represent a promising direction to better understand the microbiome. However, our effort is limited in

both volume and breath of the data that is used for training the microbial language model. Currently, our pre-trained model is primarily optimized for tasks involving human gut microbiomes based on 16S data. Despite this, the utility of our model extends beyond its initial configuration. With adjustments, our methodology can be highly versatile, offering numerous paths for generalization.

Specifically, it is possible to adapt our pre-trained language model directly to other sources and types of microbiome data, such as taxonomic profiles of Metagenome Assembled Genomes (MAGs), by replacing the initial embedding layer with one that is fine-tuned using the new source of data. Strong precedents in natural language processing support the feasibility of this approach, where pretrained models from one domain have been shown to lead to predictable transfer when adapted to another domain (e.g., from Python code to natural text [52], or from natural text to image classification [53]). Finally, we are enthusiastic about the potential to develop a unified model by training on a broad spectrum of microbiome data, encompassing various sources and modalities, to create a generalized, versatile microbiome model capable of instantaneous adaptation to the varied data distributions encountered in different studies and methodologies across the microbiome research landscape.

## Supporting information

**S1 Fig. Vocabulary and contextualized embedding spaces colored by different levels of the phylogenetic hierarchy: phylum, class, order, and family.**
(EPS)

**S1 Table. Top 10 non-disease associated ASVs.** Entries match those in Table 5.
(CSV)

**S2 Table. Top 10 disease associated ASVs.** Entries match those in Table 6.
(CSV)

## Author contributions

**Conceptualization:** Xiaoli Fern.

**Data curation:** Christine Tataru, Maude M. David.

**Formal analysis:** Quintin Pope.

**Funding acquisition:** Maude M. David, Xiaoli Fern.

**Investigation:** Quintin Pope, Rohan Varma.

**Methodology:** Quintin Pope, Rohan Varma, Xiaoli Fern.

**Project administration:** Xiaoli Fern.

**Software:** Quintin Pope, Rohan Varma, Christine Tataru, Maude M. David.

**Supervision:** Xiaoli Fern.

**Validation:** Quintin Pope, Rohan Varma, Maude M. David, Xiaoli Fern.

**Visualization:** Quintin Pope.

**Writing – original draft:** Quintin Pope, Maude M. David, Xiaoli Fern.

**Writing – review & editing:** Quintin Pope, Maude M. David, Xiaoli Fern.

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
