## [Decision Letter · Decision Letter 0]

30 Nov 2023

Dear Mr Pope,

Thank you very much for submitting your manuscript "Learning a deep language model for microbiomes: the power of large scale unlabeled microbiome data" for consideration at PLOS Computational Biology.

As with all papers reviewed by the journal, your manuscript was reviewed by members of the editorial board and by several independent reviewers. In light of the reviews (below this email), we would like to invite the resubmission of a significantly-revised version that takes into account the reviewers' comments.

The reviewers were overall positive about the manuscript, but all reviewers indicate a need for additional explanation and justification for the architecture and the training methods used. These critiques should be addressed to improve the clarity and utility of the manuscript.

We cannot make any decision about publication until we have seen the revised manuscript and your response to the reviewers' comments. Your revised manuscript is also likely to be sent to reviewers for further evaluation.

Sincerely,

Nic Vega, Ph.D.

Academic Editor

PLOS Computational Biology

Stacey Finley

Section Editor

PLOS Computational Biology

The reviewers were overall positive about the manuscript, but all reviewers indicate a need for additional explanation and justification for the architecture and the training methods used. These critiques should be addressed to improve the clarity and utility of the manuscript.

Reviewer's Responses to Questions

**Comments to the Authors:**

Reviewer #1: 1. The authors gave NLP examples when introducing the microbiome language model (e.g. Line 96 at Page 7). The authors should consider examples for microbiome instead.

2. Why did the authors used absolute positional embeddings and why did the authors freeze the embedding layer when training?

3. Does the multi-head attention have special meaning in microbiome language model?

4. The authors used masked taxa prediction task for pre-training the microbiome language model. Is there any other pretext task for pretraining microbiome language model, e.g. next sentence prediction in NLP?

5. The baseline method should be a method that widely used in the relevant field. The comparison experiments are not very convincing.

6. The performance show in Table 3 is for the ensemble methods. What is performance of the original model? Why did the authors use ensemble of Transformer as their method while this approach is seldom used in NLP field.

7. Would the limitation of 512 tokens in Transformer affect the method proposed?

8. Some sentences are confusing. For example, Line 256 at page 14, “Given a machine learning model …”

Reviewer #2: This paper presents novel methods for classifying unlabeled microbiome data utilizing a transformer-based architecture's contextualized representation to enhance classification of microbiome data. The paper reads well overall and the methods and evaluations mostly clear. However, there are a few minor points:

1. The transformer models is referred to in a few places (ie line 100) as being a recent development from NLP. The wording of this is awkward. Since have been around since 2017 now and hardly recent in NLP, and Transformers are not models, but architectures and have multiple configurations - I would refer to transformer-based architectures/models throughout. You might consider introducing the ELECTRA-based model earlier, which is a transformer-based architecture to help clarify.

2. The manuscript states that the 'CLS' token serves to "summarize the full sample" and is critical for classification. This might be a slight misrepresentation in the context of ELECTRA, which isn't designed intrinsically to use the 'CLS' token for summarization or classification. It might be more accurate to state that in your specific application, you utilize the 'CLS' token's embeddings for these tasks, as a downstream decision. (149-152, 187-188)

3. If AUROC and AUPR are chosen to the exclusion of other metrics, a justification would add rigor to your manuscript. This could be based on the nature of your data, the specific challenges of your prediction tasks, or other relevant factors. (ie why not report F1, Recall, Precision, etc)

Reviewer #3: The authors present the first transformer model trained on microbial taxonomic profiles. They not only tested the generalizability of their model embeddings on various tasks and external validation cohorts but also attempted to interpret what kind of information these embeddings capture. However, the authors omitted some technical details on data handling and model training and did not justify model architecture choices. Moreover, I suggest the authors use other state-of-the-art ML methods as the baseline for comparison in addition to the one they established in their previous study. Finally, the model interpretation subsection needs substantial reworking as it is somewhat inconclusive in its current form.

Below I outline my main concerns in more detail, which, if addressed, would strengthen the manuscript:

1) While the authors provide an extensive description of the transformer training procedure, I find the other technical details somewhat lacking. For example, the authors mentioned that they rank order taxa and embed the top 512 species without any particular justification. Or, they use GloVe embeddings from their previous study, but it would be nice to provide a brief description for the readers not familiar with that work. Also, the dimensionalities of the embedding and projection space look somewhat arbitrary, so I wonder if the authors did any hyperparameter tuning or used any other intuition to pick these numbers. I would also add some of these details in Fig 1.

2) The same is true for the details of model pretraining and fine-tuning. What were the stopping criteria? Did the model loss or task accuracy reach saturation? It would also be useful for the readers to have a sense of how much time it took to train this model, as well as how hard it will be to fine-tune it for a specific dataset.

3) I would also suggest the authors describe their choice of baseline for model performance comparison instead of plainly citing their previous work. While looking at this subsection, I was wondering how their method will compare to some other state of the art ML approaches like SIAMCAT, DeepMicro, DeepBioGen (another way to produce “robust” embeddings), PopPhy-CNN, or MetAML. I think, it would be great if authors can demonstrate that this computationally intense LLM training was worth it in terms of accuracy performance. If their embeddings are indeed as generalizable as they claim, this will be a more convincing argument than the current tables 2 and 3. 

4) I agree with the authors that Fig 4. clearly demonstrated that transformer training improved taxa embeddings. I would recommend to add cluster purity as a sub panel to this figure instead of hiding it in the supplementary. Additionally, I suggest the authors color this figure according to other more fine-grained taxonomic levels (e.g. by genus as in the previous study). It would be nice to see whether these clusters correspond to specific family/genera or not.

5) I appreciate that the authors tried to interpret what their model learned. However, I find this subsection somewhat lengthy and unclear. For example, in the IBD task, I suggest the authors provide some statistics on how generalizable across the datasets are the most contributing features. The authors mentioned that they “validated” taxon attribution if it has the same sign in all tree cohorts, so I wonder how many biomarker taxa might be added/excluded if we add yet another external cohort to this analysis. I also suggest that instead of citing and discussing all the previous anecdotal evidence the authors compile some sort of statistics of how many known or perhaps unknown IBD markers their model discovered. For example, there are lists of IBD-associated taxa listed in various databases (e.g.: https://gmrepo.humangut.info/phenotypes/comparisons/D006262/D015212). Or one can compare the taxa identified using LLM embeddings and the list of marker taxa derived from the same dataset by commonly used statistical approaches.

6) Since I went back and read the previous paper, I have been wondering whether the authors can make a similar argument about the “context” of embeddings as they did for GloVe embeddings. In the previous study, they showed that this context seems to be driven by metabolic functions, which makes a lot of sense from the microbial ecology standpoint. It would be interesting to see if there is any functional signal here as well. And in this line of thought it would be nice to see whether the same attribution analysis for the dietary task will discover taxa harboring specific genes. I believe a finding like this will strengthen the paper's conclusions.

Below are some of my minor concerns on the manuscript:

1) line 74 “how …” grammar should be corrected.

2) I suggest the authors discuss how hard it would be to use their embeddings outside of 16S data (i.e. for WGS-derived taxonomic profiles).

**Have the authors made all data and (if applicable) computational code underlying the findings in their manuscript fully available?**

Reviewer #1: Yes

Reviewer #2: Yes

Reviewer #3: Yes

PLOS authors have the option to publish the peer review history of their article (what does this mean?). If published, this will include your full peer review and any attached files.

Reviewer #1: No

Reviewer #2: No

Reviewer #3: No
---

## [Decision Letter · Decision Letter 1]

9 Jun 2024

Dear Mr Pope,

Thank you very much for submitting your manuscript "Learning a deep language model for microbiomes: the power of large scale unlabeled microbiome data" for consideration at PLOS Computational Biology.

As with all papers reviewed by the journal, your manuscript was reviewed by members of the editorial board and by several independent reviewers. In light of the reviews (below this email), we would like to invite the resubmission of a significantly-revised version that takes into account the reviewers' comments.

The reviewers indicate that the current manuscript is substantially improved; however, a number of points remain, which may require clarification rather than further work. In particular, the reviews of the revised manuscript indicate a need for clarification on some points of data properties and data handling, as well as a need for improved documentation of the provided code.

We cannot make any decision about publication until we have seen the revised manuscript and your response to the reviewers' comments. Your revised manuscript is also likely to be sent to reviewers for further evaluation.

Sincerely,

Nic Vega, Ph.D.

Academic Editor

PLOS Computational Biology

Stacey Finley

Section Editor

PLOS Computational Biology

The reviewers indicate that the current manuscript is substantially improved; however, a number of points remain, which may require clarification rather than further work. In particular, the reviews of the revised manuscript indicate a need for clarification on some points of data properties and data handling, as well as a need for improved documentation of the provided code.

Reviewer's Responses to Questions

**Comments to the Authors:**

Reviewer #1: The authors have addressed most of the reviewer’s comments. Here are a few minor points for this manuscript.

1. Please highlight the best performance in Table 3. For those baselines proposed by other research works, please add the corresponding citations.

2. The authors perform data augmentation by randomly deleting 10% of the input taxa in each training sample to reduce overfitting. Could the authors give citations to this technique or explain more details?

Reviewer #3: The authors have addressed most of my concerns and the manuscript has been significantly improved.

Reviewer #4: # Major Comments

The manuscript presents a novel application of transformer models to microbiome data, demonstrating the potential of leveraging large-scale, unlabeled microbiome datasets to develop robust predictive models. The authors state that their approach shows improved performance in multiple prediction tasks, such as predicting Irritable Bowel Disease (IBD) and dietary patterns, which highlights the effectiveness of contextualized taxa representations. Additionally, the authors provide model interpretations in order to derive biological insight from their methodology.

While I find this work to be novel and potentially impactful, I have many major and minor concerns that I believe should be addressed in order make the manuscript suitable for publication in Plos Computational Biology.

## Dataset Split

It appears that the authors applied GloVe on the entire American Gut Project (AGP) dataset and subsequently used a subset of this AGP dataset for testing (see minor comments below). If true, this approach appears to constitute data leakage, given that GloVe is assessing taxon co-occurrance across all samples, even those in the AGP used for model testing. Such data leakage would thus likely produce an overly optimistic estimation of model performance and generalization.

Moreover, I believe that at least for the Halverson dataset, there are multiple samples per individual, so the samples are not independent. It’s not clear to me whether part of the Halverson dataset was used for fine-tuning, while the other part was used for valiation (clarification in this regard in the Methods would be helpful). If this is indeed the case, did the authors split the dataset so that all samples belonging to the same individual were solely in the train (fine tune) or test split? The same goes for the other datasets; for instance, the AGP definitely includes multiple samples per individual, but it is not clear if the authors blocked by individual when creating the train/test splits.

## Dataset Skew

The Halverson and HMP2 datasets used in this study are highly skewed towards positive classifications for IBD, with 90% and 79% of observations being IBD positive, respectively. This imbalance caps the error rates resulting from false positive classifications and potentially skews the performance metrics. The authors should consider addressing this imbalance, perhaps through techniques such as oversampling the minority class, to ensure that the model performance is not artificially inflated by the dataset skew.

## Methods

The methods section lacks crucial details about the “metabolic pathways” analysis presented in the results. For instance, the paper does not specify how correlations were calculated or the number of ASVs included in the analysis. Additionally, the criteria used for the permutation test to filter correlations are not clearly defined. Providing these details is necessary for understanding the robustness and validity of the findings presented.

## Code

First, it is a bit troubling that the code was not made directly available, and apparently was not assessed in the first round of reviews.

Second, the code provided by the authors lacks adequate documentation, making it difficult for other researchers to reproduce the work. While the README does describe each file in detail, there is no documentation on how one would utilize the files and code to reproduce the work (e.g., train the model or evaluate the model) or apply the authors’ model to novel datasets. Moreover, the code itself generally lacks documentation (e.g., function-level documentation on the purpose of the function, input parameters, and output). Comprehensive documentation is essential for ensuring that the research can be replicated and that the developed tools can be effectively applied to new research goals. The lack of clear instructions and comments in the codebase detracts from the overall impact of this work.

## Computation

The authors do not state the amount of computation required for training and inference. This information is crucial in evaluating the feasbility of the authors’ approach, especially when scaled to larger datasts.

# Minor Comments

Introduction: The repeated use of “bacteria” in the Abstract and Introduction is misleading, as the dataset includes both bacteria and archaea (at least, there is not mention of filtering out archaeal ASVs). The authors should ensure that the archaeal component is appropriately acknowledged.

Line 44: The statement “enables contextualized interpretation of individual bacterial species” is misleading since ASVs are not equivalent to species taxonomic units, as defined by standardized taxonomies such as GreenGenes, SILVA, or GTDB. The same goes for any futher mention of ASVs as “species”.

Line 77: The analogy comparing microbiome data to textual sentences is flawed since microbiome samples do not have an inherent ordering like syntax in sentences. This analogy should be revised or omitted.

Line 174: Typographical error, “..” should be corrected to “.”.

Line 176: Clarification is needed on which samples from the AGP dataset were used to train the GloVe model. If samples used for training were also included in the test splits, data leakage has occurred. According to Tataru and David, 2020: “By applying the GloVe algorithm to 18,480 gut microbiome samples from the American Gut Project…”. Thus, it appears that GloVe was trained on all 18480 AGP samples used in this work.

Line 235: The term “each taxa” should be corrected to “each taxon.”

Line 273: Clarify whether fine-tuning was performed only with the AGP dataset or on each dataset in the study.

Line 289: Reiterate that ASVs are not equivalent to microbial species as defined by established taxonomies.

Line 308: State whether all samples are from different individuals or if there are repeated observations. If there are repeated observations, clarify whether individuals were split between training and validation datasets.

Line 329: Confirm the presence of repeated observations in the Halfvarson dataset and detail the distribution of these repeated observations and how non-independence was handled during model fine-tuning.

Line 432: Quantify the correspondence between ASV distances in embedding and taxonomic space to prevent incorrect conclusions from t-SNE or UMAP projections.

Line 479: It is unclear if the embeddings significantly differ; neither Figure 6 (heatmap) nor Figure 7 (histogram) convincingly demonstrate this.

Line 488: “each taxa” should be corrected to “each taxon” or “each ASV” for accuracy.

Line 496: The statement’s validity is questionable due to potential artifacts from t-SNE and UMAP. Consider a more quantitative approach, such as clustering embeddings and assessing the distribution of attribution values among clusters.

Tables 4 & 5: Provide ASV identifiers and/or 16S sequences so that researchers can utilize this information for future work.

Figure 6: Specify the measure of correlation used.

**Have the authors made all data and (if applicable) computational code underlying the findings in their manuscript fully available?**

Reviewer #1: Yes

Reviewer #3: Yes

Reviewer #4: Yes

PLOS authors have the option to publish the peer review history of their article (what does this mean?). If published, this will include your full peer review and any attached files.

Reviewer #1: No

Reviewer #3: No

Reviewer #4: No
---

## [Decision Letter · Decision Letter 2]

5 Dec 2024

PCOMPBIOL-D-23-01135R2

Learning a deep language model for microbiomes: the power of large scale unlabeled microbiome data

PLOS Computational Biology

Dear Dr. Pope,

Thank you for submitting your manuscript to PLOS Computational Biology. After careful consideration, we feel that it has merit but does not fully meet PLOS Computational Biology's publication criteria as it currently stands. Therefore, we invite you to submit a revised version of the manuscript that addresses the points raised during the review process.

Please submit your revised manuscript within 60 days Feb 04 2025 11:59PM. If you will need more time than this to complete your revisions, please reply to this message or contact the journal office at ploscompbiol@plos.org. Please include the following items when submitting your revised manuscript:

We look forward to receiving your revised manuscript.

Kind regards,

Stacey D. Finley, Ph.D.

Section Editor

PLOS Computational Biology

Stacey Finley

Section Editor

PLOS Computational Biology

Feilim Mac Gabhann

Editor-in-Chief

PLOS Computational Biology

Jason Papin

Editor-in-Chief

PLOS Computational Biology

**Additional Editor Comments:**

The issues raised by the reviewer warrant serious consideration and should be addressed in a revised manuscript.

**Journal Requirements:**

1) Please ensure that all table files have corresponding citations within the manuscript .Currently, tables 7 and 8 are not cited within the manuscript. Please include the in-text citations of the tables.

2) We notice that your supplementary Figure is included in the manuscript file. Please remove it and upload it with the file type 'Supporting Information'. Please ensure that each Supporting Information file has a legend listed in the manuscript after the references list.

3) Please amend your detailed Financial Disclosure statement. This is published with the article. It must therefore be completed in full sentences and contain the exact wording you wish to be published.

1) State the initials, alongside each funding source, of each author to receive each grant. For example: "This work was supported by the National Institutes of Health (####### to AM; ###### to CJ) and the National Science Foundation (###### to AM).".

If you did not receive any funding for this study, please simply state: The authors received no specific funding for this work.

**Reviewers' comments:**

Reviewer's Responses to Questions

Reviewer #1: The authors have addressed all the comments. It is suggested to accept this manuscript for publication

Reviewer #4: # Major comments

While Pope, Varma et al., have definitely improved the manuscript, I find the manuscript still lacking in a few areas, which are listed below. I believe that these items should be addressed prior to publishing the article.

## Dataset split

I thank the authors for clarifying. I should note that randomly spliting the dataset, instead of using cross-fold validation can easily result in less reliable performance metrics. However, the dataset is somewhat large, so this potential issue is less of a concern.

## Dataset Skew

Given that AUC focus on TPR vs. FPR, 90% true positives can be misleading, since the FPR will be calculated from a small number of negative instance (DOI:10.1145/1143844.1143874).

The authors state: “… we do not want to have to re-adapt our model to new class skews during testing on novel datasets.”; however, real-world datasets will likely substantially differ in label skews (e.g., datasets with a low number of true positives, unlike the Halverson and HMP2 datasets). The authors have not shown whether their model can generalize to datasets with low prevalences of true positives. This detracts substantially from the impact of the work.

More generally, the authors did not directly compare their model to standard microbiome machine learning baselines. The other models tested were dimension reduction or deep learning approaches, but there was no comparison to statistical machine learning models (e.g., linear, random forest, or SVM), trained directly on microbiome features (e.g., ASVs or ASVs aggregated at the genus or family taxonomic level). Literature such as https://www.biorxiv.org/content/10.1101/2024.09.16.613342v2,
https://doi.org/10.1038/s41467-022-34405-3, and https://doi.org/10.3389/fmicb.2023.1261889 has shown that simple models can often outperform substantially more complicated deep learning approaches, in the domain of biology when dataset size is limited. Given this literature, and the lack luster performance of the authors’ model versus the other models assessed (Table 4), I am skeptical on whether the authors’ model would out-perform a simple random forest model trained on taxon abundances (or presence-absence).

The authors could argue that their approach is more interpretable than random forest or other statistical ML models; however, I find the authors’ demonstration of interpretabliy to lack rigor. The description of Fig 5 includes an abundance of “seems to be” and “appear to” phrases. More generally, the interpretive ability of t-SNE and UMAP have been called into question (DOI:10.1038/s41467-019-13056-x, DOI:10.1038/nbt.4314); thus, it is unclear whether the following descriptions in the manuscript of the t-SNE ordinations are useful: “appear to lie on long ‘strands’” and “made up of smaller ‘threads’ very close together”. I previously asked for more quantitative approaches to interpreting the results (e.g., clustering or comparing taxonomic distance to embedding distance), but the authors have not included such analyses.

## Methods

I thank the authors for clarifying the methods. I stil do not find the heatmaps in Fig 6 to be useful; it is quite hard to discern a qualitative difference. The same applies to Fig 7, in that the distributions are quite similar. While the K-S and E-S tests show low P-values, what is the effect size? Moreover, I believe both tests assume independent data points, which seems to be violated for the embedding data.

## Code

While the authors state that they have provided their code, I could not find it. The Sept 19, 2024 repository of files associated with this work includes the following files: ensemble.zip, microbiomedata.zip, pretrainedmodels.zip, README.md, seqs_.07_embed.fasta, vocab_embeddings.npy. The “## File Structures:” section of the README lists 3 zip files that are missing from Dryad: DeepMicro.zip, property_pathway_correlations.zip, microbiome_transformers.zip. According to the “## File Structures:” section, these are the zip files that contain all of the code (R, Python, Bash, and Jupyter Notebooks).

The authors should provide all of their code, and the code should be well-organized so that the work can be fully replicated.

# Minor comments

Line 101: "a bacteria" => "a bacterium”

Line 518: ”By comparing the observed correlations to this null distribution, we filtered out correlations that were not statistically significant” => what is “statistically significant” in this context?

**Have the authors made all data and (if applicable) computational code underlying the findings in their manuscript fully available?**

Reviewer #1: None

Reviewer #4: **No: **Code missing

PLOS authors have the option to publish the peer review history of their article (what does this mean?). If published, this will include your full peer review and any attached files.

Reviewer #1: **Yes: **Shankai Yan

Reviewer #4: No

**Figure resubmission:**
---

## [Editor Report · Decision Letter 3]

24 Mar 2025

Dear Mr Pope,

We are pleased to inform you that your manuscript 'Learning a deep language model for microbiomes: the power of large scale unlabeled microbiome data' has been provisionally accepted for publication in PLOS Computational Biology.

Best regards,

Stacey D. Finley, Ph.D.

Section Editor

PLOS Computational Biology

---

## [Editor Report · Acceptance letter]

PCOMPBIOL-D-23-01135R3

Learning a deep language model for microbiomes: the power of large scale unlabeled microbiome data

Dear Dr Pope,

I am pleased to inform you that your manuscript has been formally accepted for publication in PLOS Computational Biology. Your manuscript is now with our production department and you will be notified of the publication date in due course.

With kind regards,

Lilla Horvath
